

# On the impact of non-Gaussian wind statistics on wind turbines - an experimental approach

Jannik Schottler[1], Nico Reinke[1], Agnieszka Hölling[1], Jonathan Whale[2], Joachim Peinke[1], and Michael Hölling[1]

[1]ForWind, University of Oldenburg, Institute of Physics, Küpkersweg 70, 26129 Oldenburg, Germany
[2]Murdoch University, School of Engineering and Information Technology, Murdoch, WA, 6150 Australia

*Correspondence to:* Jannik Schottler (jannik.schottler@forwind.de)

**Abstract.** The effect of intermittent[1] and Gaussian inflow conditions on wind energy converters is studied experimentally. Two different flow situations were created in a wind tunnel using an active grid. Both flows exhibit nearly equal mean velocity values and turbulence intensities, but strongly differ in their two point statistics, namely their distribution of velocity increments on a variety of time scales, one being Gaussian distributed, the other one being strongly intermittent. A horizontal axis model wind turbine is exposed to both flows, isolating the effect of the differences not captured by mean values and turbulence intensities on the turbine. Thrust, torque and power data were recorded and analyzed, showing that the model turbine does not smooth out intermittency. Intermittent inflow is converted to similarly intermittent turbine data on all scales considered, reaching down to sub-rotor scales in space, indicating that it is not correct to assume a smoothing of wind speed fluctuations below the size of the rotor.

## 1 Introduction

Wind energy converters (WECs) work in a turbulent environment and are therefore turbulence driven systems. The dynamic wind interacts with the *system dynamics*, resulting in the output parameters of a wind energy converter system such as power, mechanical loads or other quantities of interest.

Generally, the characteristics of the output dynamics of a WEC need to be understood in detail for multiple reasons. Power fluctuations have been reported in numerous studies, causing challenges in grid stability (e.g. Chen and Spooner, 2001; Sørensen et al., 2007; Carrasco et al., 2006). Drive train and gearbox failure rates remain high, adding to the cost of energy since gearboxes are among the most expensive parts of WECs. These types of failures are likely to be linked to torque fluctuations (e.g. Musial et al., 2007; Feng et al., 2013). Next, turbulent wind affects extreme and fatigue loads, which is clearly related to the lifetime of WECs (Burton et al., 2001).

Wind dynamics in the atmospheric boundary layer have been investigated extensively. Numerous studies report on non-Gaussian characteristics of wind speed fluctuations, see (e.g. Boettcher et al., 2003; Morales et al., 2012; Wächter et al., 2012; Liu et al., 2010). Further, findings of non-Gaussian wind statistics have been implemented in simulations by a variety of

---

[1]It should be noted that, throughout this paper, *intermittency* refers to a non-Gaussian, heavy-tailed distribution of *increments* as defined in Sec. 2.



methods, see (e.g. Nielsen et al., 2007; Gong and Chen, 2014; Mücke et al., 2011).

In the field of wind energy research, it is still unclear to what extent wind dynamics transfer to the parameters of a WEC such as loads, power etc. Most likely, this depends on the relevant time scales, which change with the system dynamics. Therewith, the conversion from wind to power, loads etc. vary with the turbine type. Consequently, it is of importance what scales in time

and space are relevant to quantify the impact of turbulent wind on WECs (van Kuik et al., 2016) and scale dependent analyses become necessary.

Mücke et al. (2011) found that intermittent inflow conditions do not affect rain flow distributions of the torque significantly. However, similarly intermittent torque fluctuations based on a numeric wind turbine model used in FAST in combination with AeroDyn (Moriarty and Hansen, 2005) were found. Gong and Chen (2014) investigated the short and long term extreme re-

sponse distributions of a wind turbine during Gaussian and non-Gaussian inflow conditions using the aeroelastic tool FAST (Jonkman and Buhl Jr, 2005). The extreme turbine responses to non-Gaussian inflow were considerably larger than the ones to Gaussian wind. However, Berg et al. (2016) recently reported a vanishing effect of non-Gaussian turbulence on extreme and fatigue loads based on large-eddy simulation (LES) generated wind fields in combination with aeroelastic load simulations using HAWC2 (Larsen and Hansen, 2007). It was concluded that non-Gaussianity in sub-rotor size eddies are filtered by the rotor. In

contrast, Milan et al. (2013) showed that, based on field data, multi-MW WECs convert intermittent wind speeds to turbulent like, intermittent power with fluctuations down to the scales of seconds. Even on the scale of an entire wind farm, intermittent power output was reported. To summarize, different simulations and data from real turbines deliver an inconclusive answer on our posed question on the conversion from turbulent inflow to wind turbine data. It is not clear to what extent non-Gaussian flow conditions transfer to turbine data.

With the present work we contribute wind tunnel experiments to the ongoing discussion on the conversion process of non-Gaussian wind statistics to wind turbine data such as power, thrust and torque. A model wind turbine and an active grid for flow manipulation were used in order to examine to what extent Gaussian distributed and highly intermittent wind speeds affect the model turbine dynamics differently.

This paper is organized as follows: Sec. 2 gives an overview of commonly used methods to characterize wind speed time

series, parts of which are applied to offshore measurement data and simulated wind speed time series. Mathematical tools used throughout this paper are introduced here. Next, Sec. 3 describes the experimental methods used, including the setup, the definition of examined quantities and their processing. Sec. 4 shows the results of the experiments, which are discussed in Sec. 5. Finally, Sec. 6 gives the conclusion of the findings.

## 2 Atmospheric flows

As WECs work in turbulent wind conditions, a proper characterization of these conditions becomes necessary (van Kuik et al., 2016). The industry standard IEC 61400-1 defines procedures for wind field description (International Electrotechnical Commission, 2005). Ten minute mean values and turbulence intensities are considered along with power spectral densities. Therewith, only the first two statistical moments of a velocity time series are taken into account.



In this section, we give a brief overview of the methods used in the industry standard and beyond, along with their mathematical background, without claims of completeness. We refer to Morales et al. (2012) for a more detailed elaboration.

A general first step to characterize a time series of wind velocities, $u(t)$, is the definition of velocity fluctuations (Burton et al., 2001),

$$u'(t) = u(t) - \langle u \rangle, \tag{1}$$

where $\langle u \rangle$ denotes the mean value of $u(t)$. A commonly used quantification of the general level of turbulence is the turbulence intensity (TI),

$$\mathrm{TI} = \frac{\sigma_{\widetilde{T}}}{\langle u \rangle_{\widetilde{T}}}, \tag{2}$$

with $\sigma_{\widetilde{T}}$ being the standard deviation of $u(t)$ during the time $\widetilde{T}$ (Burton et al., 2001). Accordingly, $\langle u \rangle_{\widetilde{T}}$ denotes the mean value over the same time span, which is typically ten minutes in industry standards. Notice, since $\sqrt{\langle u'^2(t) \rangle_{\widetilde{T}}} = \sigma_{\widetilde{T}}$, only the first two statistical moments of the one point quantity $u'$ are considered when describing a velocity time series by its fluctuations and/or turbulence intensity as previously defined.

Going one step further in the sense of two point quantities, we will consider velocity changes during a time lag $\tau$ and refer to them as velocity *increments*,

$$u_\tau = u(t + \tau) - u(t) \tag{3}$$

throughout this paper. The $n^{\mathrm{th}}$ order moments of $u_\tau$ are commonly referred to as $n^{\mathrm{th}}$ order structure functions (Wächter et al., 2012). The second order structure function,

$$\langle (u(t + \tau) - u(t))^2 \rangle, \tag{4}$$

is directly linked to the autocorrelation function $R_{u'u'}(\tau)$, see (Morales et al., 2012) for details. The autocorrelation function of $u'(t)$ and $u'(t + \tau)$ is connected to the power spectral density (PSD) by the Fourier transformation[2]. Therewith, the PSD, which is used broadly in wind field models such as the well-known Kaimal model (Kaimal et al., 1972), comprises the same information as the second order structure function.

In order to include *all* higher order structure functions, $\langle u_\tau^n \rangle$, we will consider the probability density functions (PDF) of velocity increments, $p(u_\tau)$, for different time lags $\tau$ and refer to them as *increment PDF*. We normalize $u_\tau$ by its standard deviation $\sigma_\tau$ for better visual comparison.

For design load calculations, different turbulence models are used. One, which is suggested by the IEC standard, is the Kaimal model, which considers power spectral densities and features merely Gaussian statistics. In this paper, we investigate to what extent wind characteristics not captured by standard models impact wind turbines. For further instance, we consider a synthetic wind speed time series based on the Kaimal turbulence model, created using the software TurbSim (Jonkman, 2009)

---

[2] $\mathscr{F}\{R_{u'u'}(\tau)\} = S(f)$, with $\sigma_{u'}^2 = \int S(f) \mathrm{d}f$ and $S(f)$ being the power spectral density (Press et al., 1992).



and compare it to offshore wind speed measurements, taken from the FINO1 platform at 80 m height. 10 Hz data of one year were considered and ten minute records of $7\,\mathrm{m\,s^{-1}} \leq \langle u(t)\rangle_{10\mathrm{min}} \leq 8\,\mathrm{m\,s^{-1}}$ were selected. The approximately 3700 records were then combined and used in this analysis, in order to ensure close-to stationary conditions. Tab. 1 shows the mean values, standard deviations and turbulence intensities of both data sets. As can be seen, the synthetic time series and the field measure-

| time series | $\langle u\rangle$ [m s$^{-1}$] | $\sigma_u$ [m s$^{-1}$] | TI [%] |
|---|---|---|---|
| Kaimal | 7.51 | 0.54 | 7.21 |
| FINO1 | 7.50 | 0.54 | 7.18 |

**Table 1.** First two statistical moments and turbulence intensities of a synthetic wind speed time series based on the Kaimal model and offshore data (FINO1). Values are rounded to two decimal places.

ments are very similar regarding their mean values and turbulence intensities (cf. Tab. 1). Going further, Fig. 1 shows $\mathrm{p(u_\tau)}$ of both data sets, showing distinct differences regarding their distributions of increments. The Kaimal model comprised purely Gaussian statistics, while the offshore data feature intermittent increment distributions. As shown in Tab. 1 and Fig. 1, certain characteristics of a wind speed time series, extreme events in particular, are not reflected correctly using standard methods. In this paper, we elaborate if, and to what extend flow characteristics that are *not* captured by the standards (e.g. the first two

statistical moments) impact wind turbines. We follow an experimental approach using a model wind turbine in a wind tunnel equipped with an active grid, allowing the generation of various turbulent inflow conditions. By tuning the intermittency while preserving mean wind speeds and turbulence intensities, the effect of intermittency is isolated.

## 3  Methods

### 3.1  Experimental setup

**Wind tunnel and active grid**

The experiments were conducted in a wind tunnel of the University of Oldenburg in open jet configuration. The outlet of $0.8\,\mathrm{m} \times 1\,\mathrm{m}$ (height $\times$ width) was equipped with an active grid for turbulence generation as described by Reinke et al. (2016). The grid is made of nine vertical and seven horizontal axes with square metal plates attached. 16 stepper motors allow an individual motion of the axes and thus flow manipulation. However, throughout the experiments, all axes were excited simul-

taneously. We define a flap angle $\alpha$, whereas $\alpha = 0°$ is in alignment with the main flow direction (*open*) and $\pm 90°$ corresponds to maximum blockage, respectively. At $\alpha = 0°$, the blockage of the grid is approx. $6\%$, considering the cross sectional area of the grid in relation to the wind tunnel outlet (Reinke et al., 2016).

The excitation protocols of the motors were designed so that two different flow situations with the same mean wind velocities and comparable turbulence intensities were realized. At the same time, they strongly differ in their distributions of increments:

one flow (A) being Gaussian distributed, the other one (B) being highly intermittent on a broad range of time scales, showing



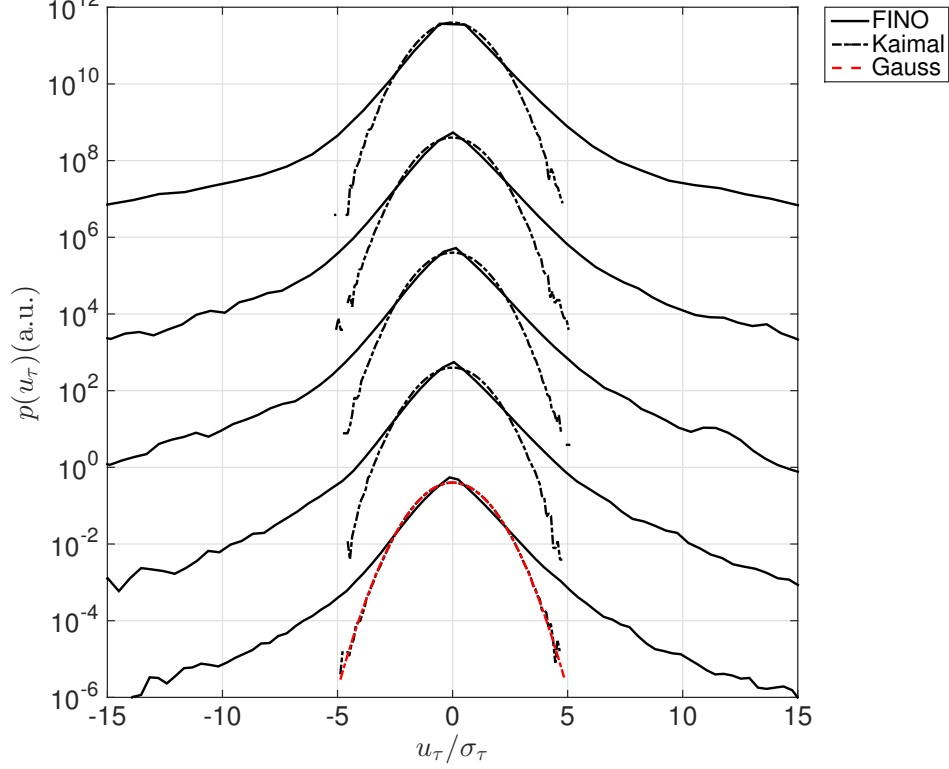

**Figure 1.** $p(u_\tau)$ for data sets based on the Kaimal model (black dashed line) and for offshore measurements, conditioned so that $\langle u \rangle = 7.5 \pm 0.5\,\mathrm{m\,s}^{-1}$ (black solid). The PDF for each scale are shifted vertically for better comparison, which is done throughout this paper. Scales from top to bottom $\tau = \{60\,\mathrm{s},\ 30\,\mathrm{s},\ 10\,\mathrm{s},\ 5\,\mathrm{s},\ 1\,\mathrm{s}\}$.

a distinctly heavy-tailed distribution of velocity increments. The resulting time series are discussed in Sec. 4.1. The excitation protocol resulting in the intermittent flow features an *active* flow modulation, where $\alpha$ was changed appropriately at a maximal rate of 50 Hz. For the Gaussian flow, the axes were not moved dynamically, so that $\dot{\alpha} = 0°$.

The flows considered were characterized using three 1D hot wire probes simultaneously in one plane normal to the main flow

5    direction. The probes were arranged so that one is located at the position of the model wind turbine's hub, which was installed after flow characterization. The other two probes were positioned in 0.6 D distance in vertical and horizontal displacement at the same stream wise position, $D = 0.58\,\mathrm{m}$ being the rotor diameter of the model turbine. The hot wires are 1.25 mm long with a diameter of 5 μm. A constant temperature anemometry (CTA) module (*Dantec 9054N0802*) with a built-in low pass filter set to 5 kHz was used. Data were recorded at 10 kHz for 25 minutes using a *National Instruments cRIO-9074* real time controller

10   with in-house built LabView software. When analyzing the flows, spatially averaged mean values of the three simultaneous measurements,

$$u(t) = \frac{1}{3} \sum_{i=1}^{3} u_i(t)\,, \tag{5}$$



are considered, where the index $i$ denotes the respective hot wire. This approach is more appropriate to describe the wind speed affecting the rotor than a single point measurement.

The distance from the active grid to the rotor and hence the hot wires was 1.1 m, which was set as a compromise between two aspects: first, the further away from the outlet, the greater the influence of the emerging shear layer becomes (Mathieu and Scott, 2000), which should be limited; second, the interaction of the rotor's blockage with the active grid increases with smaller distances. Also, the evolution of the turbulence intensity and intermittency was found to decay constantly around 1 m behind the grid (Weitemeyer et al., 2013). Consequently, a distance of 1.1 m was chosen to balance the described effects.

**Model wind turbine**

A three bladed, horizontal axis model wind turbine with a rotor diameter of $D = 0.58$ m was used. The vacuum-casted rotor blades are based on a SD7003 airfoil profile; further details on the turbine design are described by Schottler et al. (2016); for details about the blade design, see (Odemark, 2012). We consider the electrical power,

$$P = P_{el} = U_{gen} \cdot I, \tag{6}$$

where $U_{gen}$ is the generator voltage and $I$ is the electric current of the circuit. $I$ is obtained by measuring the voltage drop $U_{sh}$ across a shunt resistor of $R_{sh} = 0.1\,\Omega$, so that Eq. (6) becomes

$$P = U_{gen} \cdot \frac{U_{sh}}{0.1\,\Omega} . \tag{7}$$

According to the generator's specifications, the torque $T$ is proportional to the electric current $I$,

$$T = k \cdot I, \tag{8}$$

with $k = 79.9\,\mathrm{mN\,A^{-1}}$. The turbine features an automatic load control, with the process variable of the controller being the tip speed ratio (TSR) based on hub height wind speed measurements using a hot wire probe $2/3\,D$ upstream of the rotor, cf. Fig. 2. The generator's load is controlled using an external voltage applied to a field-effect transistor (FET) within the electric circuit. Throughout this study, the TSR was set to $\lambda_{set} = 7$, based on $u_\infty = 7\,\mathrm{m\,s^{-1}}$ to ensure a stable point of operation (not in stall) during the experiments.

To measure the thrust force acting on the turbine, it was placed on a three component force balance (*ME-Meßsysteme K3D120-50* N). Only the thrust force in main flow direction is considered, thus

$$F = F_{thrust,x} . \tag{9}$$

The setup is sketched in Fig. 2; Fig. 3 shows a photograph. As shown in Fig. 2, three hot wires were installed upstream of the rotor during turbine operation. In contrast to the flow characterization, only the center hot wire signal at hub height was used when comparing inflow data to turbine data done in Sec. 4.2.




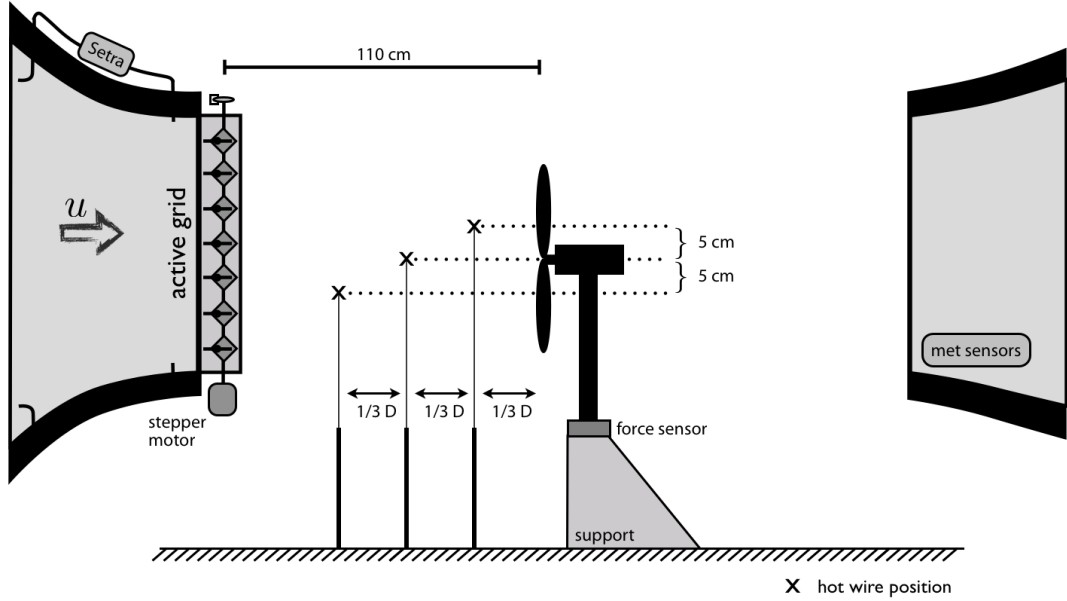

**Figure 2.** Schematic drawing of the experimental setup, side view. Scales do not match, $D = 0.58$ m.

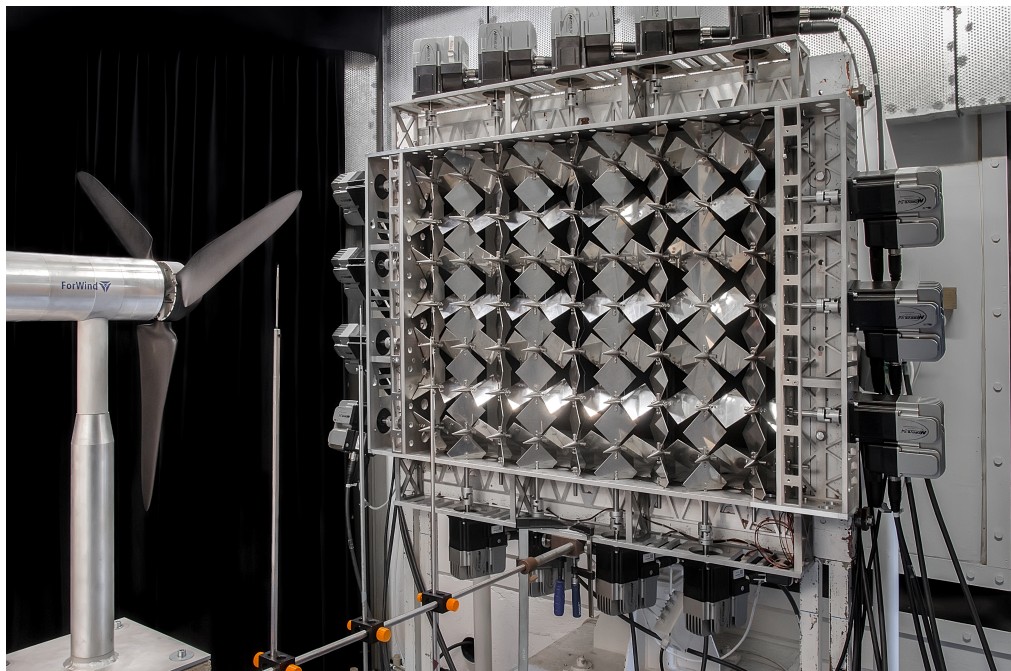

**Figure 3.** The model wind turbine and the active grid installed in a wind tunnel of the University of Oldenburg.



## 3.2 Data processing

For each experiment, data were recorded simultaneously. During flow characterization, the three hot wire probes were synchronized and during turbine data acquisition, the thrust force, power, torque and the hot wire signals were recorded synchronously. Generally, all data sets are superimposed with some kind of measurement noise, which we generally want to exclude from our analysis, while preserving the fluctuations of the turbine signals resulting from the inflow. As we analyze different parameters, an appropriate filtering of the different raw signals should, nonetheless, allow a comparison of their statistics. To begin with, $u(t)$ during the intermittent inflow B is filtered using a $6^{th}$ order Butterworth low pass filter. The cut off frequency is set to $2\,kHz$, since high frequency noise, which is typical for hot wire anemometers (Jørgensen and Hammer, 1999), should be filtered. Further, the resolved length scales corresponding to $2\,kHz$ ($\sim mm$, using Taylor's hypothesis (Mathieu and Scott, 2000)) are reasonably small for our purposes. Fig. 4 shows the PSD of the intermittent inflow (B) based on raw and filtered data. As

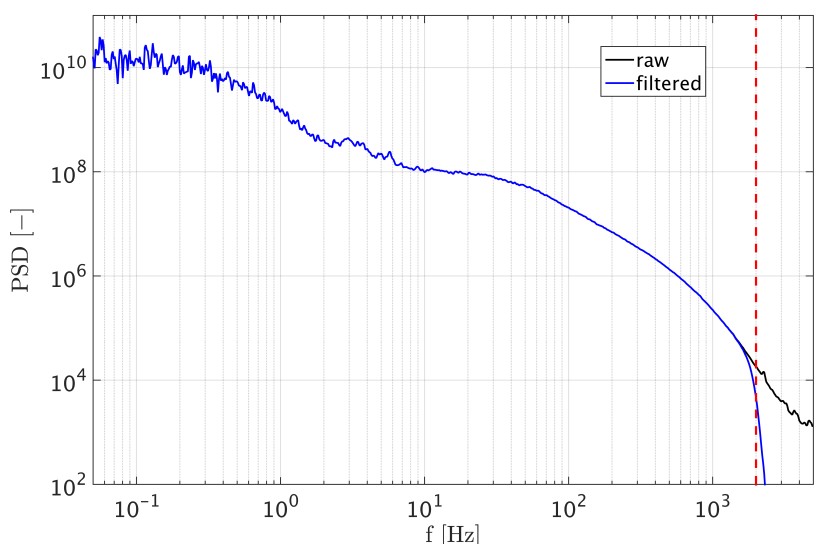

**Figure 4.** Power spectral density (PSD) of $u(t)$ for the intermittent inflow B. Raw data shown in black, filtered data with a $6^{th}$ order Butterworth lowpass filter at $f_{cut} = 2\,kHz$ shown in blue. The red dashed line marks $f_{cut} = 2\,kHz$.

we want to concentrate on the fluctuations of turbine data caused by the inflow, we estimate a maximal frequency for which the fluctuations of the respective turbine data are coherent with the fluctuations of the filtered velocity signal. Therefore, we consider the magnitude-squared coherence,

$$\gamma^2_{u'x'} = \frac{|P_{u'x'}(f)|^2}{P_{u'u'}(f)P_{x'x'}(f)} \; , \tag{10}$$

of the filtered wind speed fluctuations and the fluctuations of the respective turbine quantity $x'$ (Carter et al., 1973), with x being the power, torque or thrust respectively. $P_{u'x'}$ denotes the cross spectral density; $P_{u'u'}$ and $P_{x'x'}$ the auto spectrum. The results are shown in Fig. 5. At the values indicated by the red dashed lines in Fig. 5, the coherence of the signals is lost almost





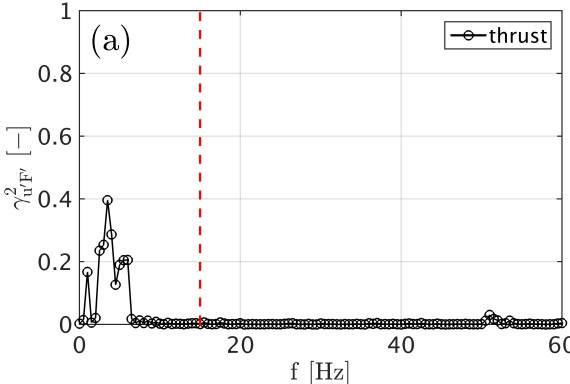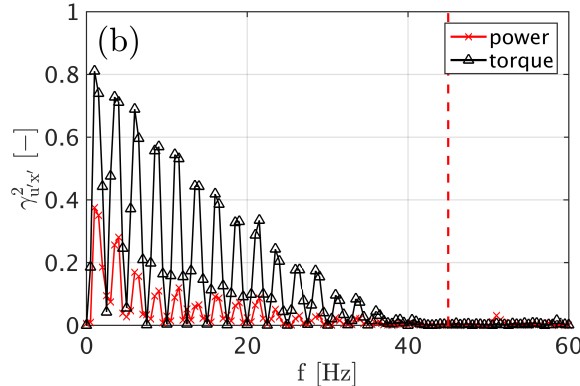

**Figure 5.** Magnitude-squared coherence of filtered hot wire data and thrust (a) as well as power and torque (b) respectively. 500 Hanning windows with 50 % overlap were used here, as suggested by Carter et al. (1973).

completely. Therefore, we chose a cut off frequency of 15 Hz for the thrust data and 45 Hz for the power and torque data to filter the raw data using a $6^{th}$ order Butterworth low pass filter. Hereby, higher frequencies are excluded, as only fluctuations resulting from the inflow should be considered. Fig. 6 shows examples of the time series of the four different signals, filtered and unfiltered. Only the filtered data sets are used for further analysis.



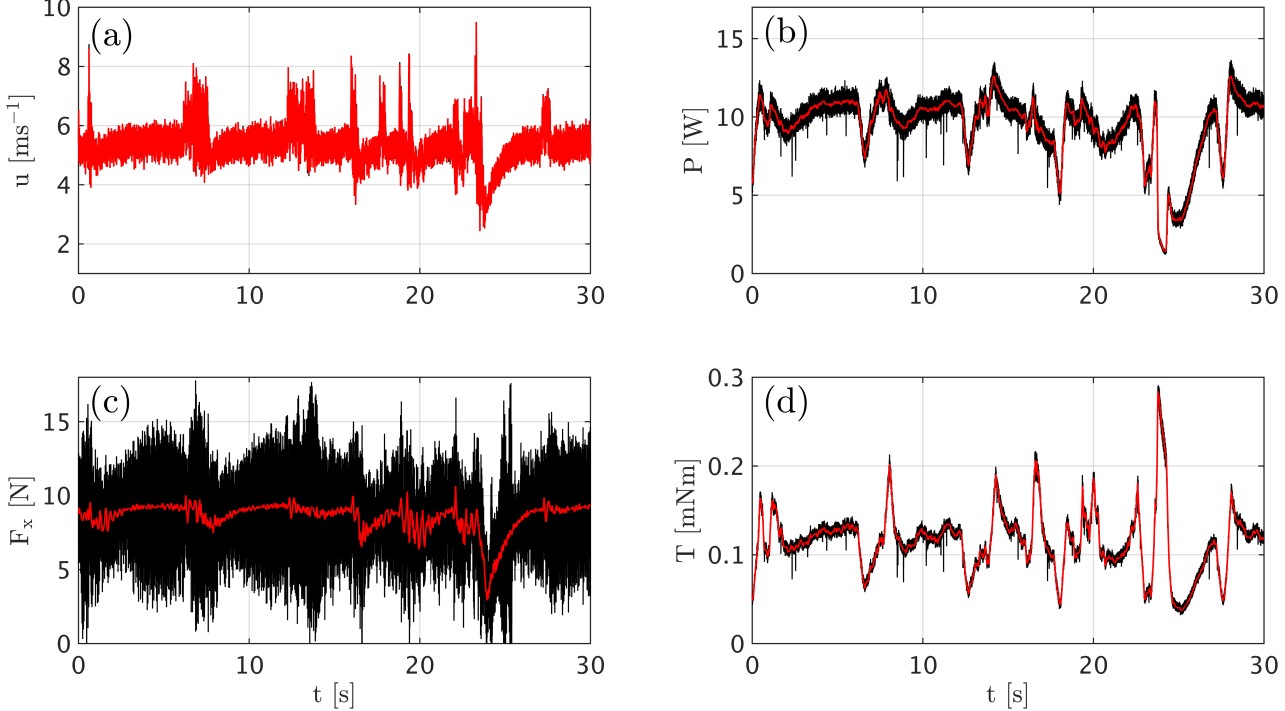

**Figure 6.** Original (black) and filtered (red) exemplary time series of the wind speed (a), power (b), thrust force (c) and torque (d). The wind speed was filtered using a $6^{\text{th}}$ order Butterworth lowpass filter at $2\,\text{kHz}$. In a similar way, the power and torque signals were filtered at $45\,\text{Hz}$ and the thrust force at $15\,\text{Hz}$.

### 3.3 Choice of scales

As previously described, we will consider increment PDF of different time scales, $p(u_\tau)$. Defining relevant scales for WECs is not trivial and is the subject of discussion throughout the research community (van Kuik et al., 2016). Therefore, a broad spectrum of time scales was chosen, ranging from the order of seconds to the smallest scales possible while applying the described filtering. By using Taylor's hypothesis of frozen turbulence (Mathieu and Scott, 2000), the chosen time scales are related to length scales of the model turbine, with $\langle u \rangle \approx 7\,\text{m s}^{-1}$. The largest scale considered is $\tau = 2\,\text{s}$, which corresponds to approximately $14\,\text{m}$ and is thus larger than the test section of the wind tunnel. Smaller time scales are based on turbine lengths and the filter frequencies, respectively. Tab. 2 gives an overview of the different scales considered. When analyzing thrust data, the smallest time scale, $\tau = 25\,\text{ms}$, was excluded due to the filtering described in Sec. 3.2.





| physical object | - | rotor diameter | - | order of blade length |
|---|---|---|---|---|
| time [s] | 2 | 0.08 | 0.067 | 0.025 |
| length/D [-] | $\approx 24$ | 1 | $\approx 0.8$ | 0.3 |
| frequency [Hz] | 0.5 | 13 | 15 | 40 |

**Table 2.** Overview of scales considered in relation to certain characteristic turbine lengths. Taylor's hypothesis is used to transfer from time to space with $\langle u \rangle \approx 7\,\mathrm{m\,s^{-1}}$.

## 4 Results

### 4.1 Inflow

Throughout the following analysis, two different, purposely created flow situations will be considered and used as inflow conditions for the model wind turbine. Fig. 7 shows the two wind speed time series as defined in Eq. (5) with $\langle u(t) \rangle \pm \sigma_{u(t)}$ indicated. Additionally, Tab. 3 lists the mean values, standard deviations and turbulence intensities for the two cases and Fig. 8 shows a 30 s excerpt. We refer to the time series as inflow A and inflow B, according to Fig. 7. It is noteworthy that in describing the wind fields by their mean values and turbulence intensities, as it is widely done, both conditions, A and B, are virtually equivalent as can be seen in Tab. 3. However, just by looking at the time series, a difference becomes obvious, which

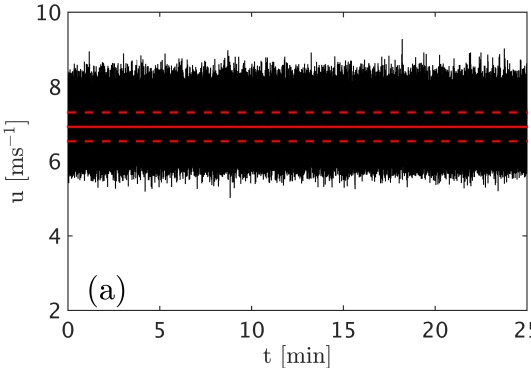
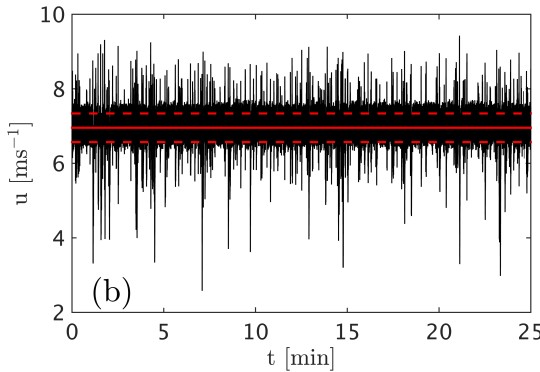

**Figure 7.** Velocity time series as defined in Eq. (5) of the two inflows considered, A and B. Further information in shown in Table 3. Solid red lines mark $\langle u(t) \rangle$ and dashed red lines indicate $\langle u(t) \rangle \pm \sigma_{u(t)}$.

| time series | $\langle u(t) \rangle$ [m s$^{-1}$] | $\sigma_{u(t)}$ [m s$^{-1}$] | TI [%] |
|---|---|---|---|
| A | 6.92 | 0.39 | 5.59 |
| B | 6.96 | 0.38 | 5.50 |

**Table 3.** First two statistical moments of the time series shown in Figure 7 and their turbulence intensities. Values are rounded to two decimal places.





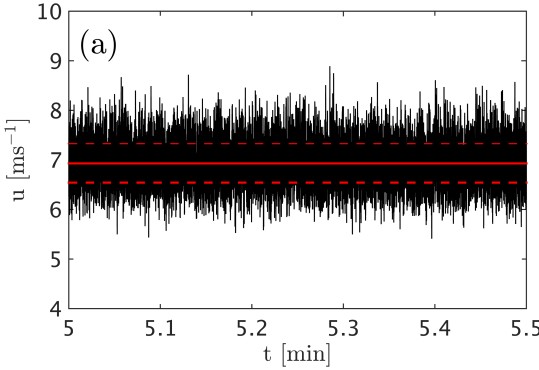
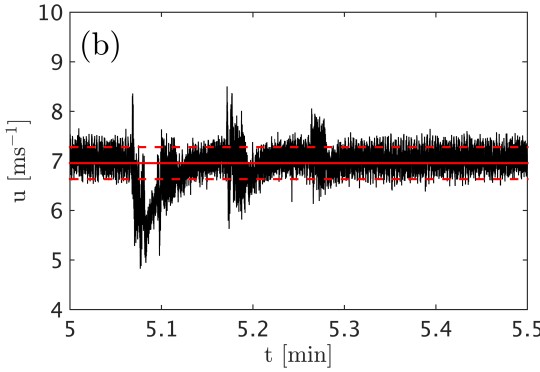

**Figure 8.** Excerpts of both time series shown in Fig. 7.

will be further investigated. Therefore, Fig. 9 shows the increment PDF $p(u_\tau)$ of both time series for the scales listed in Tab. 2. Clearly, both flows are significantly different regarding intermittency . While inflow A follows a Gaussian trend, inflow B shows a strongly heavy-tailed, highly intermittent distribution of increments. Therefore, extreme events occur significantly more frequently in inflow B as compared to inflow A. Similar discrepancies as shown in Fig. 1 for offshore measurements and simulated data become obvious.



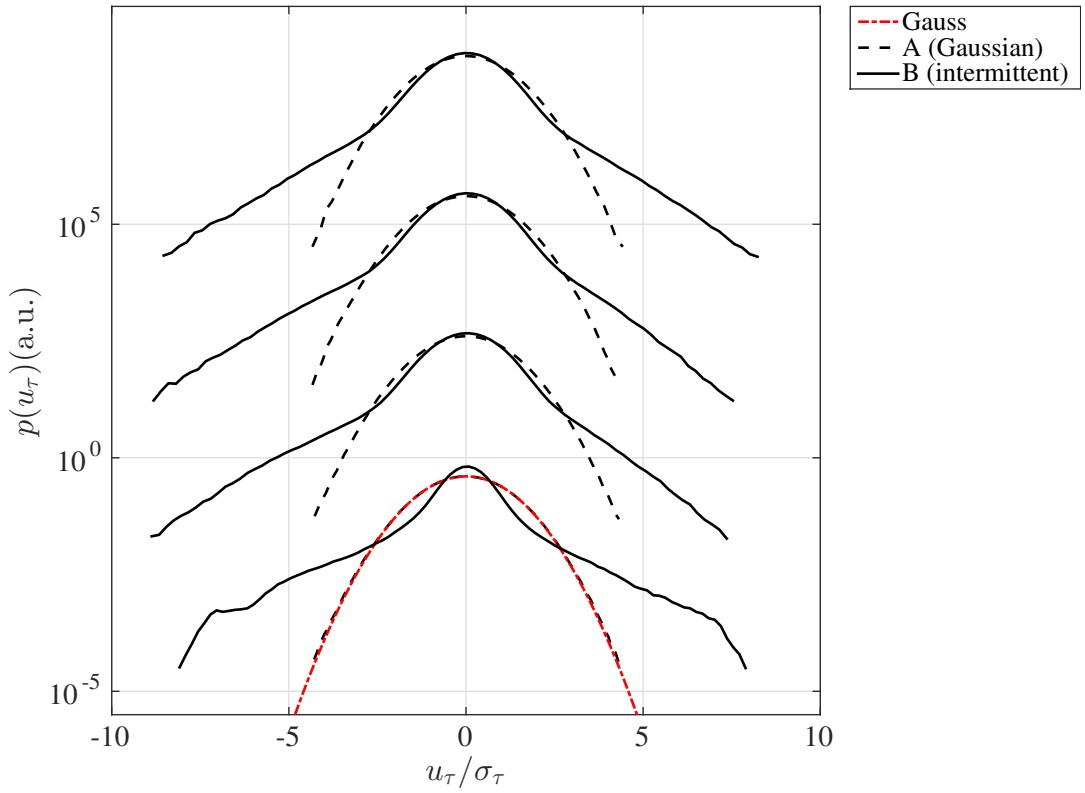

**Figure 9.** $p(u_\tau)$ of both velocity time series shown in Fig. 7, A (dashed) and B (solid), for $\tau = \{25\,\text{ms},\ 67\,\text{ms},\ 80\,\text{ms},\ 2\,\text{s}\}$ from top to bottom. The different scales are shifted vertically for presentation. A Gaussian fit (dashed red line) of $p(u_{\tau=2\,\text{s}})$ for inflow A is added to guide the eye.

### 4.2 Turbine reaction

Next, we investigate the performance data of the model wind turbine when exposed to both flows A and B. To begin with, we consider the thrust force in main flow direction, $p(F_\tau)$, in Fig. 10. Clearly, the difference between Gaussian and non-Gaussian inflow conditions remains present in the thrust data for all time scales considered. Non-Gaussian fluctuations are not filtered

5  out by the rotor. Going further, we directly compare the normalized quantities, $p(F_\tau)$ and $p(u_\tau)$, separately for both flow conditions in Fig. 11. Neither for the Gaussian nor for the intermittent case, a change in the forms of the increment PDF can be observed. Thus, we conclude that the non-Gaussian fluctuations of the inflow are not averaged out by the rotor. In Fig. 10 and 11, the smallest time scale of $\tau = 25\,\text{ms}$ is not shown for the thrust data, as that scale interferes with the previously applied low pass filter as described in Sec. 3.2.

10  So far, we have considered the thrust force of the turbine as example, showing a transfer of intermittency from $u_\tau$ to $F_\tau$ by the system dynamics of the turbine. For the power and torque we obtain similar results as for the thrust, thus we present all





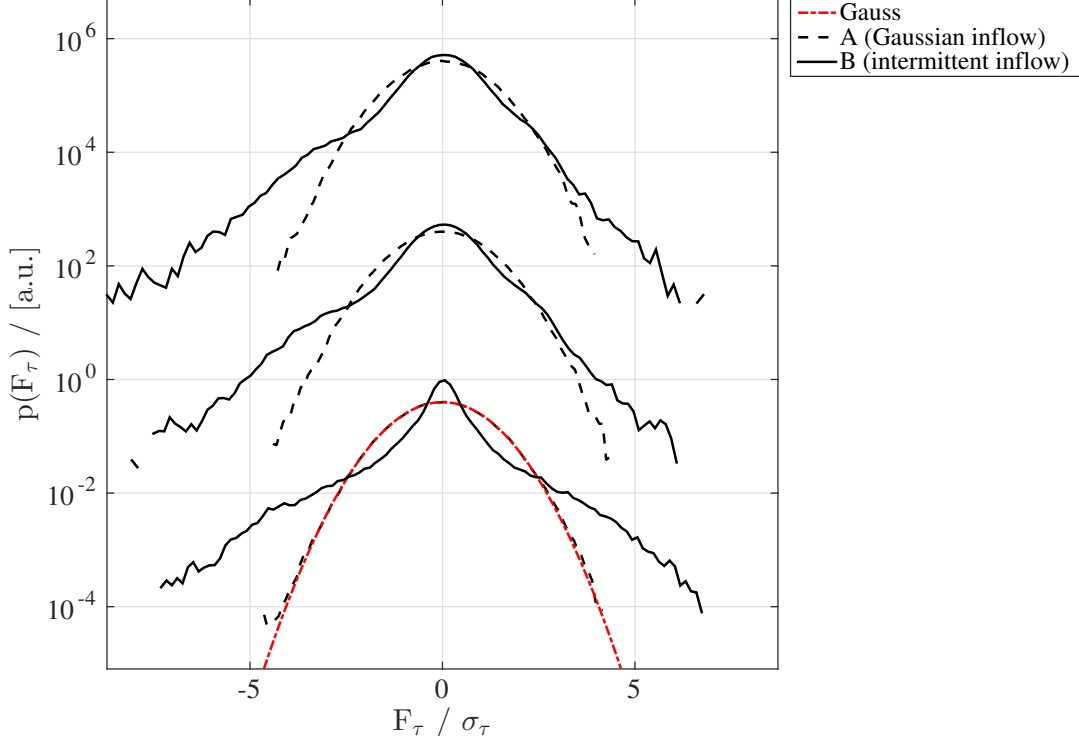

**Figure 10.** $p(F_\tau)$ of the turbine's thrust force (in main flow direction) exposed to the inflow conditions A (dashed) and B (solid), for $\tau = \{67\,\mathrm{ms}, \ 80\,\mathrm{ms}, \ 2\,\mathrm{s}\}$ from top to bottom. The different scales are shifted vertically for presentation.

quantities for the intermittent inflow together in Fig. 12. None of the quantities smooth out the intermittent inflow to a close-to Gaussian distribution. Minor deviations of the respective increment PDFs are discussed in Sec. 5.




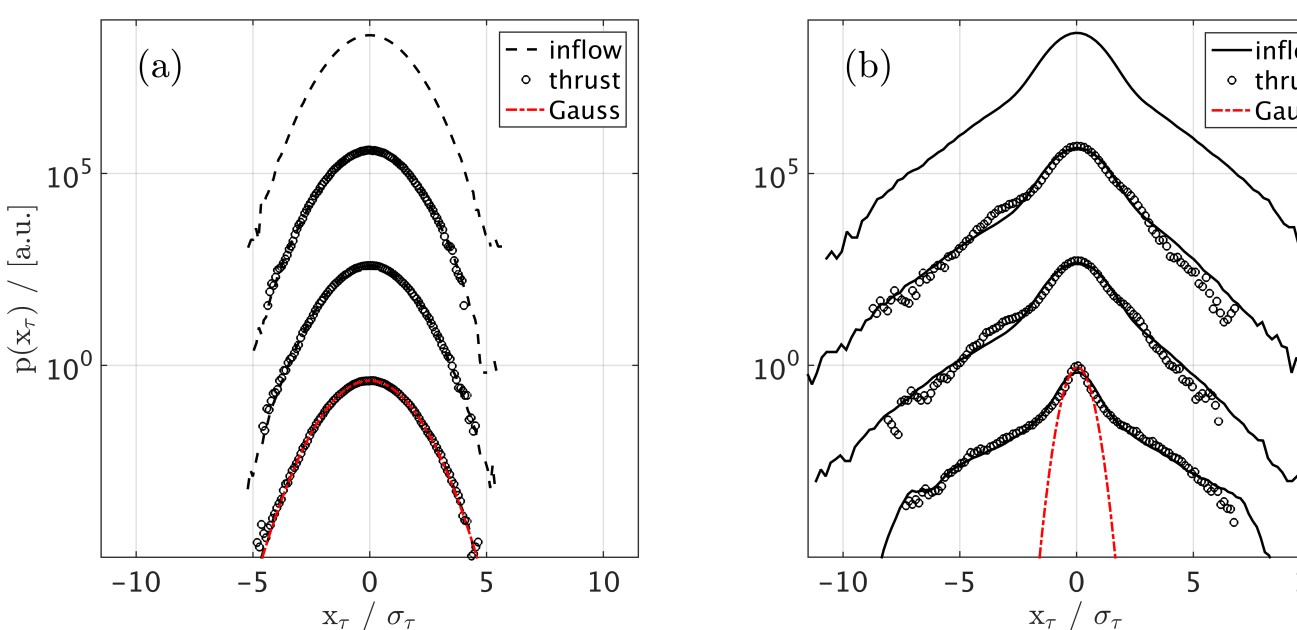

**Figure 11.** $p(u_\tau)$ (lines) and $p(F_\tau)$ (circles) for both inflow conditions Gaussian (a) and intermittent (b). Scales as in Fig. 10 from top to bottom $\tau = \{25\,\text{ms}, 67\,\text{ms}, 80\,\text{ms}, 2\,\text{s}\}$, shifted vertically for presentation.



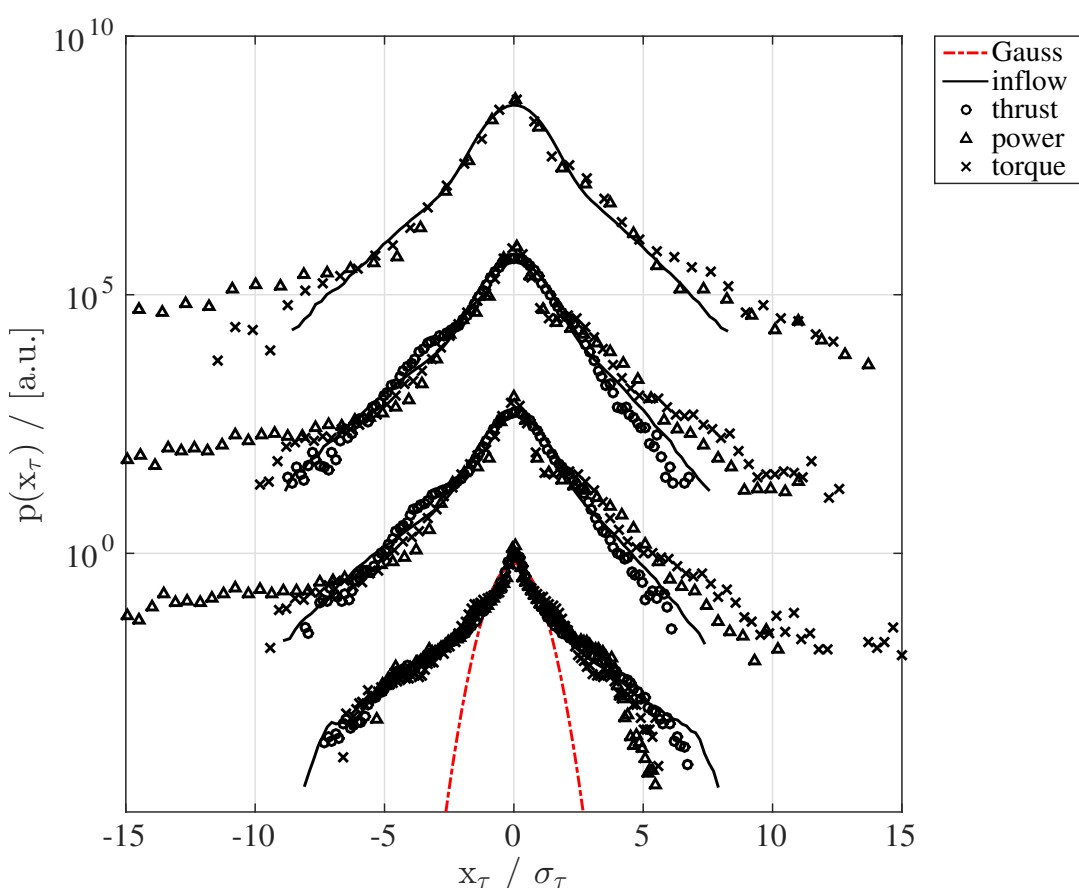

**Figure 12.** $p(x_\tau)$ for the intermittent inflow condition (line, cf. Fig. 7 (a)), thrust (circles), power (triangles) and torque (crosses). Scales as in Figure 10 from top to bottom $\tau = \{25\,\text{ms}, 67\,\text{ms}, 80\,\text{ms}, 2\,\text{s}\}$, shifted vertically for presentation.



## 5   Discussion

When processing the experimental data, signal fluctuations not resulting from the inflow are excluded from the analysis by previously applied low pass filters. While noise is only a minor issue considering the power and torque, the thrust data from the force balance is significantly superimposed by signal fluctuations resulting from the setup itself, cf. Fig. 6(c). These are most

likely arising from vibrations of the whole setup during turbine operation, ranging from the turbine itself and the support to the fixation in the ground. Those fluctuations are of an amplitude that would influence the analysis, however, they are of higher frequency than the cut off frequency of the applied low pass filter. Therefore, they are indeed excluded from the analysis. At the same time, the procedure described in Sec. 3.2 might filter fluctuations of higher frequency than the respective cut off which are actually part of what is directly related to wind speed variations. As a result, minimal time scales have to be set, potentially

excluding interesting results for smaller scales. Considering Fig. 5(a), the coherence of the hot wire signal and the thrust data is almost lost completely at approximately 10 Hz. As this corresponds to a time scale of $\tau = 0.1\,\mathrm{s}$ or a length scale of $0.7\,\mathrm{m}$ ($\approx 1.2\,\mathrm{D}$), a cutoff frequency of $15\,\mathrm{Hz}$ was chosen in order to include a scale between the rotor diameter and the blade length. Also, there might be aerodynamic effects that are of even higher frequency than the inflow fluctuations, and therefore not captured due to the filtering. As a straightforward example, a laminar flow passing a cylinder results in a well-defined frequency

due to von Kármán vortex shedding, cf. Mathieu and Scott (2000). The shedding does result from the inflow, although not being related to the fluctuations. Thus, aerodynamic effects at the rotor are possibly excluded by the low frequency filtering. Considering Fig. 12, some minor deviations between the increment PDF of the inflow and the turbine data can be observed. The torque and the power as defined in Eq. (8) and (7) are part of the electric circuit and therefore directly linked to the manipulative variable of the controller, being the voltage applied to the FET, $\mathrm{U_{FET}}$. Thus, an analysis of those quantities includes not only

fluctuations caused by the inflow, but also those resulting from the controller. As overshoots are typical for closed-loop control systems (Ogunnaike and Ray, 1994; Chien and Chung, 2003), they are much likely biasing the present analysis, especially for small time scales regarding the power and the torque. Because of that, the focus of the analysis is on the thrust data. Nonetheless, the main finding that, despite differences among the parameters, *all* quantities feature strongly intermittent distributions of increments, remains, as Fig. 12 shows.

When using the model wind turbine to grasp the impact of the different inflows considered, we do not claim full scalability. There is a Reynolds number mismatch between the scaled laboratory model and full scale turbines. Further, the model is not aero elastically scaled, which is likely to impact the scalability of the presented results. Therefore, a detailed study of the (time-) scale dependency of the results is not included here. Moreover, we concentrate on the main findings of this study, being the remaining intermittency in the turbine data on the considered scales during intermittent flow conditions.

## 6   Conclusions

In this study, an experimental setup was realized, that allows the investigation of interactions between various turbulent flows with a model wind turbine. Experiments were performed in order to elaborate on the impact of non-Gaussian wind statistics on



WECs. Our results show no filtering of the intermittent features of wind speed fluctuations found in real wind fields, which are not reflected in standard wind field descriptions, e.g. the IEC 61400-1. Intermittent inflow is converted to similarly intermittent turbine data on all scales considered, ranging down to sub-rotor scales. Thus, statistical properties of the inflow time series that are not captured by describing them by one-point statistics are of relevance and should be included in standards characterizing

5 inflow conditions. If intermittent inflows lead to intermittent loading, including extreme loads that occur much more frequent than currently modeled in the standards, then this has implications for the use of the current standards in designing wind turbines to withstand the wind conditions experienced.

## Appendix A:  Variances of increment PDF

For completeness, the variances $\sigma_\tau^2$ of every time series of increments, $x_\tau$, are shown in Tab. 4 for the synthetic and offshore date, cf. Fig. 1, and for the experimental data in Tab. 5.

| time scale $\tau$ [s] | 1 | 5 | 10 | 30 | 60 |
|---|---|---|---|---|---|
| $var(u_\tau)$, Kaimal [m$^2$ s$^{-2}$] | 0.25 | 0.47 | 0.53 | 0.58 | 0.58 |
| $var(u_\tau)$, FINO1 [m$^2$ s$^{-2}$] | 0.04 | 0.11 | 0.15 | 0.24 | 0.31 |

**Table 4.** Variances of each increment time series u$_\tau$(t), for synthetic data based on the Kaimal model and field data.

| time scale $\tau$ [s] | 0.025 | 0.067 | 0.08 | 2 |
|---|---|---|---|---|
| $var(u_\tau)$, Inflow A [m$^2$ s$^{-2}$] | 0.30 | 0.30 | 0.30 | 0.30 |
| $var(u_\tau)$, Inflow B [m$^2$ s$^{-2}$] | 0.06 | 0.08 | 0.09 | 0.252 |
| $var(F_\tau)$, Inflow A [m$^2$ s$^{-2}$] | - | 0.03 | 0.06 | 0.13 |
| $var(F_\tau)$, Inflow B [m$^2$ s$^{-2}$] | - | 0.11 | 0.14 | 0.86 |
| $var(P_\tau)$, Inflow A [m$^2$ s$^{-2}$] | 0.03 | 0.12 | 0.16 | 3.31 |
| $var(T_\tau)^*$, Inflow B [m$^2$ s$^{-2}$] | 1.17 | 6.28 | 8.48 | 133.72 |

**Table 5.** Variances of each increment time series for the experimental data. $var(u_\tau)$ corresponds to the graphs shown in Fig. 9, $var(F_\tau)$ to the graphs in Fig. 10, $var(P_\tau)$ and $var(T_\tau)$ to p($P_\tau$) and p($T_\tau$) respectively, as shown in Fig. 12.
*) $\times 10^{-5}$.



*Acknowledgements.* Parts of this work was funded by the Reiner Lemoine Stiftung (RLS). The authors thank Stefan Ivanell for providing the rotor blade design. Further, we thank Philip Rinn and Matthias Wächter for fruitful discussions. Finally, the authors thank the German Bundesamt für Seeschifffahrt und Hydrographie (Federal Maritime and Hydrographic Agency) and the DEWI for providing the FINO1 data.




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
