# Peer review of "On the impact of non-Gaussian wind statistics on wind turbines - an experimental approach"

_Wind Energy Science, 2016_

## Referee Comment (RC1) · Anonymous Referee #1 · 4 Sep 2016

The manuscript entitled "On the impact of non-Gaussian wind statistics on wind turbines – an experimental approach" deals with the experimental observation that the turbulent properties as the intermittency that characterize the flow impacting a wind turbine are transmitted to its operating properties, even for sub-rotor scales.

The manuscript content is of great interest for the wind energy sector since the outcomes can have a direct impact of the way of designing wind turbines in order to improve their robustness and durability. The obtained results were already observed through numerical simulations but it is an additional step to prove this overall transmission of intermittency at the lab scale.

On the other hand, some questions that arise through the present study on the cut-off

frequency of this transmission is not enough deeply discussed (see major comments).

Consequently, I recommend the publication of the article, considering that major corrections will be made according to the recommendations given below.

Major comments:

- P3, l10-11, Eq 5 : a space average is applied to the incoming flow measurements, arguing that "This approach is more appropriate to describe the wind speed affecting the rotor than a single point measurement". Please elaborate your argumentation.

- If this approach is more appropriate, why is it not used in the following part of the study?

- §3.2 : please show PSD for all signals and discuss them

- Fig 5 : the coherence functions for power and torque are not continuous, but show regular peaks. Why? Don't we expect continuous functions? Please elaborate an explanation.

- Wind turbine data sets are low-pass filtered with a cut-off frequency of 15Hz for thrust and 45Hz for power and torque data. These cut-off frequencies are very close to the frequencies related to the time scales of interest, 13 and 40Hz.One can therefore expect that the signal distortion due to filtering (magnitude damping and phase shift) affects the wind turbine signals at the frequencies of interest, and so their unsteady properties, including intermittency. In other words, how confident can one be in the increment PDFs obtained for the thrust with tau = 0.067s, whereas the signal is low-pass filtered at 15Hz; and for the power and torque with tau = 0.025s, whereas the signals are low-pass filtered at 45Hz?

- Fig12 : PDFs show strong asymmetric distributions for power and torque. Please add the skewness and flatness values for all signals and for both inflow conditions and comment them

[Figure]

Minor comments:

- P2, l7: "Rainflow distribution" instead of "Rain flow distribution". Please explain briefly what it is.

- P2, l8: describe FAST at the first time of appearance in the manuscript (aeroelastic tool)

- P2, l20: "we contribute THROUGH wind experiments to…"

- P3, Eq 4: add " = " to the equation

- P3, l19: write the relationship between the second-order structure function and the autocorrelation function.

- P3, l25: define sigma_tau (formula)

- P4, l7 : "As shown in Tab.1 and Fig. 1,…" : remove Tab.1

- Figures 1, 9 to 12: you choose in Fig 1 to sort the plots (top to bottom) according to a decreasing time scales. You choose the opposite for other plots. Please make all plots consistent.

- P5, l6-7: it is not clear at which downstream distance from the rotor the measurements are performed. Is it 1D?

- Eq 7 : write "R_sh" instead of "0.1 Ohm"

- P6, l21-22: please indicate the velocity rotation, the rpm and the associated frequency of the wind turbine model.

- P8, l17 : "autospectra" instead of autospectrum"

- P10, l7 and Tab.2: "turbine dimensions" instead of "turbine lengths"?

- Tab. 2 : "time scale tau" instead of "time"

- Tab 2 : if I compute the frequency related to the time scale 0.08s, I obtain 12 instead

of 13Hz

- Fig. 11: use solid lines for the inflow for both plots

[Figure]

---

## Referee Comment (RC2) · Anonymous Referee #2 · 13 Sep 2016

General comments:

This manuscript investigates the effects of non-Gaussian wind conditions on the performances of a scaled horizontal axes wind turbine. The study is experimental. It uses one set of field measurements, synthetic data created using the software TurbSim and wind tunnel measurements. The idea behind this study is very interesting. However, the study does not try to relate any of the analysis to the real atmospheric conditions. Although they show results from one filed campaign, it is not clear what the implications of their study are. That is, the authors talk about non-Guassian wind statistics without saying what kind of atmospheric flows (winds) are non-Gaussian. For instance, synoptic winds in atmosphere have Gaussian statistics.The Discussion section is more a

summary than the critical discussion of the study and its implications.

Taking into account the good objectives and motivations behind this research, on one hand, and a large number of specific comments I provided below, on the other hand, I assign moderate to major revisions for this manuscript. Please see my specific comments below.

Specific comments:

1. You should not have footnotes in Abstract. Abstract should be a stand-alone section without references to the rest of the paper.

2. P1, L11. "The dynamic wind interacts..." What is a dynamic wind? This might imply that there is a static wind, which I never heard of. Wind is movement of air, thus it is dynamic by definition. Why not saying "Wind interacts..."

3. Not sure what is your rule to italicize words. I have nothing against italicizing the important words and terms, but in your manuscript you are using it for that purpose, as well as for the names of some instruments, modules, etc. I suggest you use it only to highlight important words.

4. Citations in the text should be from oldest to the latest. For example, P1, L1 has citations that are in a random order; similarly citations at the end of P1 are also randomly listed. Please correct that throughout the text.

5. P1, L20. When discuss the non-Gaussian characteristics of wind, you should mention some of the atmospheric phenomena that create those winds; like downbursts, for example, which are quite frequent in Europe and elsewhere. Gust fronts are other phenomena associated with non-Gaussian winds. There are several papers by Giovanni Solari and his group on that subject. For instance, De Gaetano et al. (2014) demonstrated the non-Gaussianity statistics of some non-synoptic winds (see Figures 2, 3 and 4 in their paper). Papers like this would strengthen your study, as they show that there are some atmospheric phenomena that generate non-Gaussian wind statistics.

[Figure]

De Gaetano P, Repetto MP, Repetto T, Solari G. 2014. Separation and classification of extreme wind events from anemometric records. Journal of Wind Engineering and Industrial Aerodynamics 126: 132–143. DOI: 10.1016/j.jweia.2014.01.006.

6. Symbols in your equations are not the same as symbols in the text. Your u'(t) in the text does not look like u'(t) in equations. It is not italicized in the text. Please be consistent and correct these. (I gave an example of u'(t), but this holds for all of your symbols).

7. What is the reason behind using the wind speed interval between 7 m/s and 8 m/s and not some other or perhaps wider interval?

8. You jumped right away to advanced statistical techniques, i.e. structure functions without showing some basic statistics. Please plot wind speed histogram of field measurements and fit it with Gaussian distribution. Synoptic winds show high degree of Gaussianity (please see the reference I provided above and some of the papers cited in that reference). Therefore, it is strange that your filed data are highly non-Gaussian. Thus, I would like to see a histogram and PDF of field measurements. It will also demonstrate that, while wind speed distribution is (maybe) Gaussian, the wind speed increment does not have to be Gaussian. I believe that further contributes to your paper.

9. Table 1 cuts a sentence in half. Please organize the text so that you don't have these discontinuities. It decreases the readability of your manuscript.

10. P2, L78. If I am correct, you are using only the interval [7, 8] m/s. That being said, what extreme events are you referring to when you say extreme events are not reflected correctly using standard model.

11. P6, L1. Why is this spatially averaged wind speed more appropriate to describe the wind speed conditions than a single point measurements? Please explain.

12. P6, L910. This sentence has too many semi-colons. Please reformulate this

sentence in order to remove these unnecessary semi-colons.

13. P6, L27. Why did you decide to use only a single hot wire signal for the comparison in Section 4.2 and not the spatially averaged data that you used for flow characterization?

14. What are the uncertainties and errors in all your measurements (wind tunnel, filed measurements, thrust, etc.)? Uncertainties in measurements should be well documented.

15. In Figure 6, is the time series of wind speed synthetically created or is it from the wind tunnel measurements (or maybe field measurement)? Either way, that time series looks very artificial to me. Also, you said that your field measurements are in the interval [7, 8] m/s, but your wind speeds in Figure 6 are around 5 m/s. Is it due to the scaling or something else? Please explain.

16. P10, L6. What is the purpose to analyze scales that cannot be produced in your wind tunnel? That length scale cannot be replicated inside of your chamber.

17. Your Table 2 is very confusing. What does the number 0.067 represent? That is, what is the column between "rotor diameter" and "order of blade length"?

18. P11, L3. "... analysis, two different, purposely created..." Please reformulate this sentence. Sounds strange.

19. P17, L15. Vortex shedding, i.e. frequencies at which vortex shed is defined by the Strouhal number, which in turn depends on the Reynolds number. That being said, how is that shedding does not depend on fluctuations in inflow? Please elaborate. If needed, please take a look at Zdravkovich's books on flow around circular cylinders.

20. P17, L16. Based on the circular cylinder example, how did you conclude that some aerodynamic effects might be excluded due to low frequency filtering? You use a "Thus" at the beginning of that sentence and I do not see how that claim results from the previous discussion. Please explain.

21. The last sentence in the Discussion section is confusing. To me, it sounds like you are saying that the focus of this study is to analyze what is presented in the study, which is redundant. Please reformulate or explain what information you want to convey in that sentence.

22. I believe you should emphasize more on the importance of your study in the Discussion section. Try to relate your findings, at least qualitatively, with the real atmospheric conditions. Also, what would be the application of your study? When can we expect non-Gaussian velocity increments and when are they Gaussian in real atmosphere? Moreover, are they ever Gaussian? All these questions could be addressed in Introduction and/or Discussion. Providing answers to those and similar questions would greatly improve the readability and contributions of your paper.

23. P18, L1. "Our results show..." Please reformulate this sentence. Not clear what you want to say.

24. Lastly, I advise the authors to find a native English speaker to proofread the manuscript.

---

## Author Comment (AC1) · 23 Oct 2016

Author's response to Anonymous Referee #1:

The authors thank the referee very much for the time and effort to review our manuscript and for the valuable comments. Please find our responses below. The original comments are stated in **bold** while our responses follow in plain text. Thank you very much.

Major comments:

P3, 110-11, Eq 5 : a space average is applied to the incoming flow measurements, arguing that "This approach is more appropriate to describe the wind speed affecting the rotor than a single point measurement". Please elaborate your argumentation.

There are two reasons to use the space averaged wind speed to describe the inflow conditions at the rotor's position.

1. Due to the rotor's rotation, the turbine is affected by the whole wind field across the rotor swept area. A single point measurement might therefore not capture important flow characteristics affecting the rotor on other positions within the rotor swept area. In numerous recent studies a rotor effective wind speed for inflow descriptions and advanced control strategies is used to capture the actual wind speed affecting the rotor [1, 2].

Based on the data of the three hot wires, we investigated whether the space averaging shows different increment statistics compared to the single point measurements. As Fig. 1 of this reply shows, there is no significant difference regarding the intermittency of the increment PDF. The small deviations, especially of the center hot wire  $u_1$ , might be explained by the second reason for the space averaging, given below.

Figure 1:  $p(u_{\tau})$  for the three single point measurements and the respective space average.

2. As described by Reinke et al. [3] in detail, the active grid is made of numerous square metal flaps that are connected by joints equipped with streamlined support structures. Therefore, a single point measurement behind a joint is not as much affected by the movement of the flaps compared to a position behind a flap. Please refer to [3] for details. However, Fig. 1 shows that the intermittent character of the inflow is obvious for all single point measurements as well as the space average. Still, we believe that for inflow characterization a space averaging gives a more appropriate description of the flow affecting the whole rotor.

We clarified this aspect in the revised manuscript as follows:

Following the concept of a rotor effective wind speed used in [2], this approach is more appropriate to describe the wind speed affecting the rotor than a single point measurement. It should be noted that our results are hardly effected by using averaged measurements as opposed to data of the central hot wire. The distance...

**If this approach is more appropriate, why is it not used in the following part of the study?**

For inflow characterization we used space averaged data for reasons mentioned above. When comparing turbine data to wind speed data, single point hot wire measurements in front of the turbine were used so that *simultane-ously* recorded data can be compared. As we mentioned above there is not a big difference between using averaged and non averaged wind speed. There is definitely another aspect, that each hot wire in front of the turbine will create a small perturbation of the inflow, especially when mounting multiple wire in one plane. Thus we prefer to work only with one hot wire for this comparison.

**please show PSD for all signals and discuss them.**

Please find the PSDs of the remaining three signals in the figures below. The PSD of the power data, Fig. 2 of this reply, clearly shows the mean rotational speed of the turbine,  $\langle \omega \rangle \approx 25.2$  Hz, and the harmonics. Also the PSD of the thrust data (Fig. 3 of this reply) shows the rotation frequency along the the harmonics, although not as clearly as for the power data. More striking are the multiple peaks that we associate with the vibrations of the whole setup including the turbine and the support structures with ground mountings. We do not believe that adding all PSDs to the manuscript will improve quality and readability. We showed the PSD of the hot wire data because the filtered signal is the basis of the approach described in section 3.2. An explanation of the regular drops of the torque-PSD in given in the following aspect.

Figure 2: Power spectral density (PSD) of the intermittent power time series, raw (black) and filtered (blue).

Figure 3: PSD for thrust data.

---

## Author Comment (AC2) · 11 Nov 2016

Authors' response to Anonymous Referee #2:
We, the authors, are very thankful for the detailed and constructive comments and greatly appreciate the willingness to review our manuscript. Please find our responses below. The original comments are shown in **bold** and the respective answers below. Excerpts of the manuscript are shown in *italic writing*, whereas additions are written in blue and deleted parts in . Thank you very much.
Specific comments:

1. **You should not have footnotes in Abstract. Abstract should be a stand-alone section without references to the rest of the paper.**
   This will be corrected by placing the description of intermittency in section 1.

2. **P1, L11. "The dynamic wind interacts: : :" What is a dynamic wind? This might imply that there is a static wind, which I never heard of. Wind is movement of air, thus it is dynamic by definition. Why not saying "Wind interacts: : :"**
   What was meant is that the wind speed is not static. We want to stress here that the wind, which interacts with the turbine, contains fluctuations/turbulence. We changed the text accordingly:

   *The  turbulent wind interacts with the system dynamics, resulting in the output parameters of a wind energy converter system such as power, mechanical loads or other quantities of interest.*

3. **Not sure what is your rule to italicize words. I have nothing against italicizing the important words and terms, but in your manuscript you are using it for that purpose, as well as for the names of some instruments, modules, etc. I suggest you use it only to highlight important words.**
   This will be corrected in the updated manuscript and initialization will be limited to important words and phrases.

4. **Citations in the text should be from oldest to the latest. For example, P1, L1 has citations that are in a random order; similarly citations at the end of P1 are also randomly listed. Please correct that throughout the text.**
   This will be corrected in the updated manuscript.

5. **P1, L20. When discuss the non-Gaussian characteristics of wind, you should mention some of the atmospheric phenomena that create those winds; like downbursts, for example, which are quite frequent in Europe and elsewhere. Gust fronts are other phenomena associated with non-Gaussian winds. There are several papers by Giovanni Solari and his group on that subject. For instance, De Gaetano et al. (2014) demonstrated the non-Gaussianity statistics of some non-synoptic winds (see Figures 2, 3 and 4 in their paper). Papers like this would strengthen your study, as they show that there are some atmospheric phenomena that generate non-Gaussian wind statistics. De Gaetano P, Repetto MP, Repetto T, Solari G. 2014. Separation and classification of extreme wind events from anemometric records. Journal of Wind Engineering and Industrial Aerodynamics 126: 132–143. DOI: 10.1016/j.jweia.2014.01.006.**

We thankfully notice the mentioned paper and like to include it in the introduction. At the same time, a clearer separation between the analyses of velocity *values* and velocity *increments* seems necessary in the manuscript. Therefore, the introduction was updated and we hope to clearly separate the velocity values and the corresponding statistics from increments, which is the focus of this paper. Increments characterize changes of the wind speed in a given time horizon, which is for example important for loads and the control system acting on actual wind values. Please find the updated version of the introduction at the end of this reply.

6. **Symbols in your equations are not the same as symbols in the text. Your u'(t) in the text does not look like u'(t) in equations. It is not italicized in the text. Please be consistent and correct these. (I gave an example of u'(t), but this holds for all of your symbols).**
   There are indeed discrepancies, which will be corrected in the updated manuscript.

7. **What is the reason behind using the wind speed interval between 7 m/s and 8 m/s and not some other or perhaps wider interval?**

   We chose the interval $7\,\mathrm{m\,s^{-1}} \leq \langle u(t) \rangle_{10\mathrm{min}} \leq 8\,\mathrm{m\,s^{-1}}$ because typically a wind turbine is in a very stable operation in partial load conditions

during those wind speeds. We wanted to exclude wind speeds close to cut-in as well as rated power. The interval gave us more than $22 \times 10^6$ data point considering 1 year of offshore data, which is more than enough. Regarding our analysis of increment PDFs, wider intervals gave very comparable results as shown in Figure 1 of this reply. Further, it has been shown in [1], Fig.5 that such a constraint will filter out intermittency effects caused by instationary conditions on large scales and thus enables to study more properly small scale turbulence effects. Therefore, we would like to stick to the original [7,8]m/s interval as the mean value and turbulence intensity match our TurbSim simulations. This aspect is further discussed in the following remark 8).

[Figure]

Figure 1: $p(u_\tau)$ for FINO1 data based on different 10min-mean intervals.

To clarify this, we added to the manuscript (p.4, lines 1 ff.):

*$10\,\mathrm{Hz}$ data of one year were considered and ten minute records of $7\,\mathrm{m\,s^{-1}} \leq \langle u(t) \rangle_{10\mathrm{min}} \leq 8\,\mathrm{m\,s^{-1}}$ were selected. The approximately 3700 records were then combined and used in this analysis, in order to ensure close-to stationary conditions.* *It has been shown by Morales et al. [1] that such a constraint will filter out intermittency effects caused by instationary conditions on large scales and thus enables to study more properly small scale turbulence effects. It should be noted that only the mean value of one ten minute record is within $7.5 \pm 0.5\,\mathrm{m\,s^{-1}}$. During*

*this time span, samples outside of this interval are included. Tab. 1 shows...*

8. **You jumped right away to advanced statistical techniques, i.e. structure functions without showing some basic statistics. Please plot wind speed histogram of field measurements and fit it with Gaussian distribution. Synoptic winds show high degree of Gaussianity (please see the reference I provided above and some of the papers cited in that reference). Therefore, it is strange that your filed data are highly non-Gaussian. Thus, I would like to see a histogram and PDF of field measurements. It will also demonstrate that, while wind speed distribution is (maybe) Gaussian, the wind speed increment does not have to be Gaussian. I believe that further contributes to your paper.**

   This is a well known feature of stationary turbulence, u' is close a Gauss, whereas increment statistics increasingly deviate from Gaussianity [2], see also Morales et al. [1] for offshore wind data. We added such a statement in the revised paper to make this point clearer to the reader, p.3, lines 13 ff [1]:

   *Going one step further in the sense of two point quantities, we will consider velocity changes during a time lag $\tau$ and refer to them as velocity increments,*

   $$u_\tau = u(t + \tau) - u(t) \tag{1}$$

   *throughout this paper. It is important to distinguish between a statistical description of the fluctuations and the increments. In stationary turbulence, $u'(t)$ is close to a Gaussian distribution, whereas increment statistics increasingly deviate from Gauss [2], which is also shown by [1] for offshore data. The $n^{th}$ order moments...*

   It is shown in the mentioned paper that the intermittency in one-point statistics is caused by the non-stationarity of $\langle u \rangle_T$ and $\sigma_T$. Mathematically, this can be shown as done below

   $$p(u) = \int p(u|\langle u \rangle_T) \cdot p(\langle u \rangle_T) \, \mathrm{d}\langle u \rangle_T. \tag{2}$$

   While $p(u|\langle u \rangle_T)$ might be Gaussian, the term $p(\langle u \rangle_T)$ reveals the large
* * *
[1]The citation style in the revised manuscript will be consistent, e.g. (Frisch 1995)

scale instationarities and causes the non-Gaussian distribution of $p(u)$. We believe that the introduction of the manuscript should clearly state the difference in analyzing the wind speed (fluctuations) and the increments. Therefore, we changed the introduction to clarify, please find the revised version at the end of this reply. It should be clear now that only velocity increments are analyzed in our paper.

In Section 2 we focused on methods used in our analyses along with relations we found necessary to follow those approaches. As mentioned in the beginning of Sec.2, we purposely did not give a complete overview of methods to describe wind speed time series. Morales et al. [1] give a detailed description of statistical methods that are used to characterize offshore data exemplary. Therefore, we limit the description in Sec. 2 to the aspect used in our analyses (increment distributions) with referring the reader to Morales et al. for further details. In addition to the new introduction, we clarified this aspect in Sec. 2 as follows p. 3, lines 1 ff.:

*In this section, we give a brief overview of the methods used in the industry standard and beyond, along with their mathematical background, without claims of completeness. Further, the methods of data analysis used in this study are introduced. We refer to Morales et al. for a more detailed elaboration. A general first step...*

9. **Table 1 cuts a sentence in half. Please organize the text so that you don't have these discontinuities. It decreases the readability of your manuscript.**
   We will correct this issue before uploading the new manuscript, however, I think a final placement of figures and tables is still to be done due to the two-column layout of the journal publications.

10. **P2, L78. If I am correct, you are using only the interval [7, 8] m/s. That being said, what extreme events are you referring to when you say extreme events are not reflected correctly using standard model.**
    We are using the interval $7\,\mathrm{m\,s}^{-1} \leq \langle \mathrm{u(t)} \rangle_{10\mathrm{min}} \leq 8\,\mathrm{m\,s}^{-1}$, so the mean wind speed of a 10-minute block is in the interval [7,8]m/s. As shown in Figure 1 of the manuscript, we are analyzing time scales of $\tau \leq 60$s, which are relevant for the turbine dynamics. So by extreme events we are referring to extreme velocity increments of multiple standard deviations $\sigma_\tau$ on small time scales below one minute *within* a 10 minute block of $7\,\mathrm{m\,s}^{-1} \leq \langle \mathrm{u(t)} \rangle_{10\mathrm{min}} \leq 8\,\mathrm{m\,s}^{-1}$. We changed the manuscript accordingly, p.4, ll 7,8:

*As shown in Tab. 1 and Fig. 1, certain characteristics of a wind speed time series, extreme  velocity increments in particular, are not reflected correctly using standard methods. In this paper,...*

11. **P6, L1. Why is this spatially averaged wind speed more appropriate to describe the wind speed conditions than a single point measurements? Please explain.**
    Please refer to our responses to the first referee's comments as this issue is addressed there.

12. **P6, L910. This sentence has too many semi-colons. Please reformulate this sentence in order to remove these unnecessary semi-colons.**
    We reformulated this sentence to:
    *The vacuum-casted rotor blades are based on a SD7003 airfoil profile. Further details on the turbine design are described by [...]. For details about the blade design, see [...].*

13. **P6, L27. Why did you decide to use only a single hot wire signal for the comparison in Section 4.2 and not the spatially averaged data that you used for flow characterization?**
    Please refer to our responses to the first referee's comments as this issue is addressed there.

14. **What are the uncertainties and errors in all your measurements (wind tunnel, filed measurements, thrust, etc.)? Uncertainties in measurements should be well documented.**

    The offshore data is publicly available and uncertainties are well documented. For the respective anemometer, which was used in our study, the uncertainty is $\approx 3\%$ [3]. We suggest to add this information along with the respective citation to the manuscript.
    For the experimental data, we estimate the statistical error of the increment PDFs by $err \approx 1/\sqrt{n}$, where $n$ is the number of events in each bin of the respective increment. For better judgment of the statistical significants of extreme events, we mark every bin with an error $< 10\%$ ($n < 100$) with a red $\times$ as exemplary done for Fig. 11(b) of the manuscript below:

[Figure]

Figure 2: Increment PDF of inflow and thrust data. Data point with an estimated error exceeding 10 % are marked in red.

We suggest to describe this procedure and to mask values with a statistical error exceeding 10% for every increment-PDF shown in the updated manuscript,.

15. **In Figure 6, is the time series of wind speed synthetically created or is it from the wind tunnel measurements (or maybe field measurement)? Either way, that time series looks very artificial to me. Also, you said that your field measurements are in the interval [7, 8] m/s, but your wind speeds in Figure 6 are around 5 m/s. Is it due to the scaling or something else? Please explain.**

The wind speed time series shown in Figure 6 is based on hot wire measurements upstream of the turbine during the intermittent inflow created by the active grid, which will be formulated more clearly in the updated version of the manuscript p.9, lines 3 ff:

*Fig. 6 shows examples of the time series of the four different signals, filtered and unfiltered.* *The graph in Fig. 6(a) shows the wind speed during the intermittent inflow upstream of the turbine. The other*

*graphs show the simultaneously recorded signals of the turbine.*

Due to the blockage of the turbine, the wind speed shown in this figure is smaller than 7m/s, which is the approximate wind speed at the rotor's position *without* the turbine being installed, please see section 4.1.

16. **P10, L6. What is the purpose to analyze scales that cannot be produced in your wind tunnel? That length scale cannot be replicated inside of your chamber.**
    The largest *time* scale we analyze is $\tau = 2\,\mathrm{s}$. Applying Taylor's hypothesis gives a length scale of $14\,\mathrm{m}$, which is larger than the test section. However, Taylor's hypothesis gives an idea of the length scale corresponding to a time scale $\tau$. This does not mean that such a large structure is present in the test section at on point of time. Further, Knebel et al. [4] show experimentally that velocity time series with an integral length scale larger than the grid itself can be created in the wind tunnel with the active grid. We do think that it makes sense to analyze a time scale of $\tau = 2\,\mathrm{s}$. We reformulate the manuscript at p.10, lines. 6 ff:

    *The largest scale considered is $\tau = 2\,\mathrm{s}$, which corresponds to approximately 14 m and is thus larger than the test section of the wind tunnel.. Thus, the turbine experiences a flow situation corresponding to a 14 m structure in the wind field having an impact on the model turbine.*

    The interpretation of a 14-m structure being present at one time in the wind tunnel is misleading, although velocity changes in the range of seconds can be created. The reformulated section should clarify this.

17. **Your Table 2 is very confusing. What does the number 0.067 represent? That is, what is the column between "rotor diameter" and "order of blade length"?**
    In the original Table 2, each column corresponds to one of the four time scales $\tau$ considered in the increment analysis. The number 0.067 represents the time scale of $\tau = 67\,\mathrm{ms}$. We suggest to update the table for better clarity, please find it below.

|  | scale 1 | scale 2 | scale 3 | scale 4 |
|---|---|---|---|---|
| time scale $\tau$ [s] | 2 | 0.08 | 0.067 | 0.025 |
| length/D [-] | $\approx 24$ | 1 | $\approx 0.8$ | 0.3 |
| physical object [*] | - | rotor diameter | - | order of blade length |

Table 1: Overview of scales considered in relation to certain characteristic turbine lengths. The time scales $\tau$ were used the the analysis. To get an idea of the spatial dimension, Taylor's hypothesis is used to transfer from time to space with $\langle u \rangle \approx 7\,\mathrm{m\,s}^{-1}$. The obtained length scale is expressed as multiples of the rotor diameter for better comparison. The length is further related to physical objects of the turbine to get a sense of the dimensions.
[*]) The physical object relates the length scales based on Taylor's hypothesis to dimensions of the model wind turbine.

18. **P11, L3. ": : : analysis, two different, purposely created: : :"**
    **Please reformulate this sentence. Sounds strange.**
    We suggest to reformulate this sentence to:
    *Throughout the following analysis, two different flow situations will be considered and used as inflow conditions for the model wind turbine.*

19. **P17, L15. Vortex shedding, i.e. frequencies at which vortex shed is defined by the Strouhal number, which in turn depends on the Reynolds number. That being said, how is that shedding does not depend on fluctuations in inflow? Please elaborate. If needed, please take a look at Zdravkovich's books on flow around circular cylinders.**
    As mentioned, the shedding frequency is defined by the Strouhal number. The shedding also occurs during laminar inflow and does therefore not depend on the fluctuations in the inflow.

    The purpose of this example is to show that an object in the flow might experience dynamics/fluctuations that do not result from velocity fluctuations in the inflow. Such effects might occur in the experiments, however, we try to focus on the turbine dynamics that *do* result from the inflow turbulence. As this example might cause more confusion than adding completeness, we suggest to delete the specific example and reformulate as follows:

    *Also, there might be aerodynamic effects that are of even higher frequency than the inflow fluctuations, and are therefore not captured due*

*to the filtering.*  *Such effects at the rotor are possibly excluded by the low frequency filtering. This study, however, focuses on dynamics caused by the inflow turbulence.*

20. **P17, L16. Based on the circular cylinder example, how did you conclude that some aerodynamic effects might be excluded due to low frequency filtering? You use a "Thus" at the beginning of that sentence and I do not see how that claim results from the previous discussion. Please explain.**
As mentioned in the previous comment, we mention effects of higher frequency than the inflow fluctuations. This implies that those frequencies are larger than the cutoff frequency of the low pass filter, cf. Fig. 5 in the manuscript. However, as this example will be deleted as mentioned in the previous comment, we do not think that a more detailed description is needed in the manuscript.

21. **The last sentence in the Discussion section is confusing. To me, it sounds like you are saying that the focus of this study is to analyze what is presented in the study, which is redundant. Please reformulate or explain what information you want to convey in that sentence.**
The last paragraph of the discussion states that we do not claim full scalability due to the mentioned reasons. We agree that the last sentence does not really fit in here as mentioning the main findings should be done in the conclusion. We delete the last sentence of the discussion.

22. **I believe you should emphasize more on the importance of your study in the Discussion section. Try to relate your findings, at least qualitatively, with the real atmospheric conditions. Also, what would be the application of your study? When can we expect non-Gaussian velocity increments and when are they Gaussian in real atmosphere? Moreover, are they ever Gaussian? All these questions could be addressed in Introduction and/or Discussion. Providing answers to those and similar questions would greatly improve the readability and contributions of your paper.**

In this study we focus on time scales $\tau \leq 60\,\text{s}$ regarding atmospheric data. We show in Fig. 1 that those increment PDF are far from Gaussian as also shown in the cited works. Throughout this paper, we concentrate on the discrepancy between the intermittency of the data and the Gaussian assumption by the industry standards. The application is to show whether this discrepancy is relevant for wind turbines. This is stated in the conclusion. However, we agree that his should be stated more clearly. We will add to the introduction p.2, line 19.:

*It is not clear to what extent non-Gaussian flow conditions transfer to turbine data. At the same time, this is a very important aspect in the design process of wind turbines and in the wind field models used. Wrong assumptions of the conversion from turbulence characteristics to wind turbine data might lead to faulty dimensioning and problems in the integration of wind energy in the power grid.*

23. **P18, L1. "Our results show: : :" Please reformulate this sentence. Not clear what you want to say.**
    What was meant is that the intermittency in the inflow is not filtered by the turbine so that the turbine data is intermittent in a similar way. We reformulate the sentence as follows:
    *Our results  do not show any filtering of the intermittent features of wind speed fluctuations found in real wind fields by the turbine. Consequently one should be aware that wind characteristics, which are not reflected in standard wind field descriptions, e.g. the IEC 61400-1, have a significant impact on wind turbines.*

24. **Lastly, I advise the authors to find a native English speaker to proofread the manuscript.**
    The updated manuscript will be carefully proofread.